# Conjugates Containing Two and Three Trithiolato-Bridged Dinuclear Ruthenium(II)-Arene Units as In Vitro Antiparasitic and Anticancer Agents

**DOI:** 10.3390/ph13120471

**Published:** 2020-12-16

**Authors:** Valentin Studer, Nicoleta Anghel, Oksana Desiatkina, Timo Felder, Ghalia Boubaker, Yosra Amdouni, Jessica Ramseier, Martin Hungerbühler, Christoph Kempf, Johannes Thomas Heverhagen, Andrew Hemphill, Nico Ruprecht, Julien Furrer, Emilia Păunescu

**Affiliations:** 1Department of Chemistry and Biochemistry, University of Bern, Freiestrasse 3, CH-3012 Bern, Switzerland; valentin.studer@dcb.unibe.ch (V.S.); oksana.desiatkina@dcb.unibe.ch (O.D.); timofelder@hotmail.com (T.F.); 2Vetsuisse Faculty, Institute of Parasitology, University of Bern, Länggassstrasse 122, CH-3012 Bern, Switzerland; nicoleta.anghel@vetsuisse.unibe.ch (N.A.); ghalia.boubaker@vetsuisse.unibe.ch (G.B.); amdouniyosra.ay@gmail.com (Y.A.); jessica.ramseier@vetsuisse.unibe.ch (J.R.); 3Laboratoire de Parasitologie, Institution de la Recherche et de l’Enseignement Supérieur Agricoles, École Nationale de Médecine Vétérinaire de Sidi Thabet, University of Manouba, Sidi Thabet 2020, Tunisia; 4Department of BioMedical Research, Experimental Radiology, University of Bern, CH-3008 Bern, Switzerland; martin.hungerbuehler@dbmr.unibe.ch (M.H.); christoph.kempf@dbmr.unibe.ch (C.K.); johannes.heverhagen@dbmr.unibe.ch (J.T.H.); 5Department of Diagnostic, Interventional and Pediatric Radiology, Bern University Hospital, University of Bern, CH-3010 Bern, Switzerland

**Keywords:** bioactive organometallics, polynuclear ruthenium compounds, antiparasitic, anticancer

## Abstract

The synthesis, characterization, and in vitro antiparasitic and anticancer activity evaluation of new conjugates containing two and three dinuclear trithiolato-bridged ruthenium(II)-arene units are presented. Antiparasitic activity was evaluated using transgenic *Toxoplasma*
*gondii* tachyzoites constitutively expressing β-galactosidase grown in human foreskin fibroblasts (HFF). The compounds inhibited *T.*
*gondii* proliferation with IC_50_ values ranging from 90 to 539 nM, and seven derivatives displayed IC_50_ values lower than the reference compound pyrimethamine, which is currently used for treatment of toxoplasmosis. Overall, compound flexibility and size impacted on the anti-*Toxoplasma* activity. The anticancer activity of 14 compounds was assessed against cancer cell lines A2780, A2780cisR (human ovarian cisplatin sensitive and resistant), A24, (D-)A24cisPt8.0 (human lung adenocarcinoma cells wild type and cisPt resistant subline). The compounds displayed IC_50_ values ranging from 23 to 650 nM. In A2780cisR, A24 and (D-)A24cisPt8.0 cells, all compounds were considerably more cytotoxic than cisplatin, with IC_50_ values lower by two orders of magnitude. Irrespective of the nature of the connectors (alkyl/aryl) or the numbers of the di-ruthenium units (two/three), ester conjugates **6**–**10** and **20** exhibited similar antiproliferative profiles, and were more cytotoxic than amide analogues **11**–**14**, **23,** and **24**. Polynuclear conjugates with multiple trithiolato-bridged di-ruthenium(II)-arene moieties deserve further investigation.

## 1. Introduction

The recent high interest in the development of therapeutic transition metal complexes is in part driven by the clinical success of the platinum(II) anticancer drug, cisplatin (*cis*-diamminedichloroplatinum(II)). Cisplatin has become an important component in chemotherapy regimens for the treatment of ovarian, testicular, lung and bladder cancers, as well as lymphomas, myelomas and melanomas [1,2,3,4]. The use of known platinum drugs is limited by the incidence of intrinsic or acquired drug resistance [5,6,7,8] and by their high toxicity and severe side-effects [9,10,11,12]. Currently, there are six platinum drugs with marketing approval in various regions throughout the world: cisplatin, carboplatin, oxaliplatin, nedaplatin, lobaplatin, and heptaplatin. Important research efforts are directed towards the development of other platinum(II) or (IV)-based drugs that overcome these drawbacks [13,14,15,16,17]. Over the last 40 years, numerous platinum drugs have entered clinical trials, but only two (carboplatin and oxaliplatin) have gained international marketing approval, and another four have obtained regulatory approval in individual countries: heptaplatin (Korea), lobaplatin (China), miriplatin and nedaplatin (Japan) [10].

Along with platinum, other transition metals including gold, titanium, rhodium, iridium, ruthenium and osmium, as well as essential metals such as iron, copper, zinc and cobalt, inspired the development of numerous organometallic compounds as potential anticancer candidates due to their variable coordination modes, redox activity and reactivity towards biomolecules [18,19,20,21,22,23]. Parallel studies were focused on the identification of other pharmacological applications of metalorganic compounds, e.g., diagnostic tools as biomolecular and cellular probes [24,25,26] or antibacterial, antifungal, and antiparasitic therapeutics [27,28,29,30,31,32,33].

Among the numerous inorganic and organometallic compounds presenting anticancer properties, ruthenium(II) and (III) complexes were identified as some of the most promising alternatives for the replacement of platinum drugs [23,34,35,36,37,38,39,40]. For these ruthenium compounds, the oxidation state of the ruthenium atom, their solubility and stability in physiological media, greatly influence their activity against cancer cells. Three ruthenium(III) metal based complexes, NAMI-A ((ImH)[*trans*-RuCl_4_(dmso-S)(Im)], Im = imidazole) and KP1019/1339 (KP1019 = (IndH)[*trans*-RuCl_4_(Ind)_2_], Ind = indazole; KP1339 = Na[*trans*-RuCl_4_(Ind)_2_]), were subjected to clinical tests [41,42,43,44], while the most advanced ruthenium(II) complex is TLD1433 ((Ru(4,4′-dimethyl-2,2′-bipyridine)2(2-(2′,2′′:5′′,2′′′-terthiophene)-imidazo [4,5-f])]Cl_2_), a photosensitizer that has recently entered clinical trials for non-muscle invasive bladder cancer [45,46,47]. Several ruthenium(II)-arene compounds including RAPTA-C ([Ru(II)(*η^6^-p*-MeC_6_H_4_Pr*^i^*)(PTA)Cl_2_], PTA = 1,3,5-triaza-7-phosphatricyclo-[3.3.1]decane) and RM175 ([Ru(II)(*η^6^-*biphenyl)(en)(PF_6_)], en = ethylenediamine) (Figure 1) were submitted to advanced in vitro and in vivo studies [48,49,50,51,52].

This opened the door for the development of a myriad of organometallic complexes bearing the half-sandwich ruthenium(II)-arene scaffold [53,54]. Among those, symmetric and mixed cationic trithiolato-bridged dinuclear ruthenium(II)-arene compounds of general formula [(*η*^6^-*p*-MeC_6_H_4_Pr*^i^*)_2_Ru_2_(*µ*_2_-SR)_3_]^+^ [55,56] and, respectively, [(*η*^6^-*p*-MeC_6_H_4_Pr*^i^*)_2_Ru_2_(*µ*_2_-SR^1^)(*µ*_2_-SR^2^)_2_]^+^ [57] (Figure 1) have shown high cytotoxicity against human cancer cells (IC_50_ values as low as 30 nM against A2780 human ovarian cancer cells and the cisplatin-resistant variant A2780cisR) [58]. This encouraged the synthesis and evaluation of various libraries of complexes based on the trithiolato-bridged di-ruthenium scaffold in the pursuit of improved anticancer properties [55,58,59], but also opened the door for a broader scope, with the assessment of alternative biological applications such as antiparasitic activity [60,61,62].

Several symmetric complexes [(*η*^6^-*p*-MeC_6_H_4_Pr*^i^*)_2_Ru_2_(*µ*_2_-S-*p*-C_6_H_4_R)_3_]^+^ with R = *p*-Me (**A**), *p*-Bu*^t^* (**B**), *p*-OH (**C**) (Figure 1) presenting interesting in vitro cytotoxicity against cancer cells (IC_50_ values of 30 nM against A2780 and A2780cisR cells for **B**) were also assessed in mice [63,64,65]. Mixed complexes (as **D**, [(*η*^6^-*p*-MeC_6_H_4_Pr*^i^*)_2_Ru_2_(*µ*_2_-SR^1^)(*µ*_2_-SR^2^)_2_]^+^ with R^1^ = CH_2_-C_6_H_4_-*p*-Bu*^t^*, R^2^ = C_6_H_4_-*p*-OH, X = Cl^-^, Figure 1) also exhibited cytotoxicity against A2780 and A2780cisR cells in the nanomolar range [57].

Despite their unusual high potency, this class of ruthenium complexes was not selective for cancer cells and displayed a low water solubility [58]. Unlike other Ru(II)-arene complexes presenting labile chlorine or carboxylate ligands, they were stable in the presence of water, amino acids and DNA [58] and interacted weakly in a noncovalent manner with proteins, including HSA (human serum albumin), Tf (transferrin), Cytc (cytochrome c), Ub (ubiquitin) and Mb (myoglobin) [58]. Only the oxidation of cysteine (Cys) and glutathione (GSH) to form cystine and GSSG, respectively, was observed in the presence of this type of complexes [55,58], but no correlation between the in vitro cytotoxicity and the catalytic activity on the glutathione oxidation was observed [55,58]. Inductively coupled plasma mass spectrometry (ICP-MS) experiments proved that complexes **A** and **B** specifically target the mitochondrion in A2780 ovarian cancer cells, with up to 97% of the Ru content found in the mitochondrial fraction [61].

Recently it was shown that some cationic trithiolato-bridged dinuclear ruthenium(II)-arene compounds display interesting in vitro activities against two apicomplexan parasites *Toxoplasma gondii* [60] and *Neospora caninum* [61], which cause abortions and fetal malformations in humans (*T. gondii*) and animals (*T. gondii* and *N. caninum*), and against *Trypanosoma brucei* [62], the causative agent of African sleeping sickness. In addition, several compounds derived from a library of coumarin conjugates of trithiolato di-ruthenium complexes showed promising activity against *T. gondii* [66]. Thus, **A** and **B** inhibited *T. gondii* proliferation with IC_50_ values of 34 and 62 nM, respectively, and they did not affect HFF host cells at dosages of 200 µM or above, resulting in high selectivity indices. Transmission electron microscopy (TEM) detected ultrastructural alterations in the matrix of the parasite mitochondria at the early stages of treatment with **D** (X^−^ = BF_4_^−^), followed by a more pronounced destruction of tachyzoites. However, compounds **A**, **B**, or **D** applied at 250 nM did not act in a parasiticidal manner. Furthermore, complexes **A**, **B**, and **D** also inhibited *N. caninum* proliferation with low IC_50_ values of 15, 5 and 1 nM, respectively. [60,61,62] TEM also indicated that the parasite mitochondrion was the primary target, but parasiticidal activity was also not detected. Complexes **A**, **B**, and **D** applied orally in a neosporosis mouse model did not reduce parasite load in the central nervous system. These encouraging results stimulated us to develop a new library of complexes based on the trithiolato-bridged di-ruthenium scaffold and evaluate them as antiparasitic agents.

Polynuclear compounds are a relatively new and successful approach in metal-based cancer chemotherapy as exemplified by the trinuclear platinum compound **BBR 3464** [{*trans*-PtCl(NH_3_)_2_}-*μ*-{*trans*-Pt(NH_3_)_2_(H_2_N(CH_2_)_6_NH_2_)_2_}][NO_3_]_4_, (Figure 1) which was evaluated in clinical trials [67,68,69,70]. **BBR 3464** exhibited interesting biological activity and a potentially different mode of action compared to mononuclear cisplatin. The introduction of two additional metal fragments to the platinum scaffold led to a higher activity in cisplatin-resistant cell lines and tumour models for the trinuclear complex [71,72,73,74,75]. The high cytotoxicity of the polynuclear platinum compound against cisplatin resistant cells was attributed to the different type of DNA-adducts formed compared to those of cisplatin, but also to the increased number of platinum moieties [71,76,77,78].

Despite **BBR 3464** failing clinical trials due to limited response in patients [68,79], the polynuclear strategy was considered promising and an increased interest in the development of other compounds with two or more metal centres emerged. Various homo-polynuclear ruthenium, osmium and gold complexes were developed, in the quest of anticancer chemotherapeutics [78,80,81,82]. Some compounds were shown to exert their anticancer activity by different modes of action compared to established drugs, including enzyme inhibition or crosslinking of DNA or proteins [78,80,81]. Numerous polynuclear half-sandwich metal-arene (ruthenium, osmium, rhodium, iridium) complexes have been investigated for their anticancer, antibacterial or antiparasitic properties [83,84,85,86,87,88,89,90,91,92,93,94,95,96,97].

In some reports, the biological activity of polynuclear compounds was compared to that of the corresponding mononuclear analogues [88,92,95,98]. However, no general trend can be determined regarding a direct correlation between the number of the organometallic units and the measured biological properties. The results remained strongly specific to the various types of compounds and each case should be considered individually. Apart the number of the metal centres, other parameters play an important role with regard to the stability and efficiency of the polynuclear complexes: (i) the nature of the metal, (ii) the type of ligands and the coordination mode (mono- or bidentate), (iii) the length, stereochemistry and steric hindrance of the linking units, (iv) the hydrophobicity and other physico-chemical properties such as the total size and charge.

Numerous homo-polynuclear Ru(II)-arene complexes with various type of connections disposed at the level of the arene or on leg ligands were previously evaluated. For example, the concept of polynuclearity was applied to RAPTA complexes by connecting two ruthenium(II) metalorganic units at the level of the arene ligand (Figure 2) [92,93]. Flexible alkyl or polyethylene glycol (PEG) diamines, as well as 1,2-diphenylethylenediamines with controlled configuration, were chosen as linkers [92,93]. In this case, dinuclear compounds presented increased cytotoxicity compared to the corresponding mononuclear derivatives when assessed in A2780, A2780cisR and HEK293 (human embryonic kidney) cells, but no important selectivity towards cancer cells could be observed. The conformation had a strong impact on the cytotoxicity and the compounds were shown to crosslink the two dominant RAPTA-C histone sites on the acidic patch of the NCP (Nucleosome Core Particle) [93]. This type of dinuclear compounds induced aberrant chromatin condensation, due to intra-nucleosomal and inter-nucleosomal cross-linking [93].

Another example is that of a series of structurally related mono- (**G**), di- (**H**) and trinuclear (**I**) compounds containing {(*η*^6^-*p*-MeC_6_H_4_Pr*^i^*)RuCl[3-oxo-κ*O*)-2-methyl-4-pyridinonato-κ*O*4]} units (Figure 2) evaluated for cytotoxicity against various cancer cells [96,97,98]. The units in the dinuclear compounds **H** were linked via flexible alkyl chains of varying lengths and a correlation between the linker size, lipophilicity, and cytotoxicity was observed. Whilst the mononuclear **G** was inactive against A2780 cells at concentrations up to 50 μM [99], the dinuclear compound bearing the longest chain (*n* = 12) exhibited an IC_50_ value of <2 μM [96]. In addition, the dinuclear compound bearing the longest spacer (*n* = 12) displayed a different mode of binding to the rest of the series and could form a high degree of DNA–protein and DNA duplex crosslinking. The extent of DNA–protein crosslinks was shown to be dependent on the spacer length [97,100]. The number of metal centres was found to be important, with the dinuclear compound being more active than the analogous mono- (**G**) and trinuclear (**I**) compounds.

A lipophilicity-cytotoxicity correlation was also observed in a series of dinuclear ruthenium(II)-*p*-cymene (cymene = *p*-MeC_6_H_4_Pr*^i^*) complexes connected at the level of the phosphine leg ligands with flexible PEG chains (**J**, Figure 2). For this library of compounds the cytotoxicity was mildly influenced by the length of the linker, the conjugates with five and six ethylene glycol units being almost 2-fold more cytotoxic compared to the other compounds on the tested cell lines [95]. The length of the linker seemed to affect to a lesser extent the selectivity of the compound for a specific cell line [95]. Nonetheless, a distinct increase in cytotoxicity was observed for the dinuclear compounds compared with the mononuclear ruthenium control complexes **K** (Figure 2) [95].

An interesting example is that of the star-shape trinuclear ruthenium(II)-*p*-cymene complexes **L** and **M** with Schiff-base ligands, tris-2-(salicylaldimine ethyl)amine and tris-2-(2-pyridylimine ethyl)amine, for which the anticancer activities were compared to those of the corresponding mononuclear analogues **N** and **O** (Figure 3) [88]. The measured IC_50_ values for compounds **L****–****O** showed that the increase in the number of metal centres was correlated to a higher cytotoxicity against cancer cells.

The antiparasitic activity of some polynuclear organometallic compounds was also studied [83,84]. For example, polynuclear pyridyl ester (**P** and **R**) and ether (**S** and **T**) complexes (Figure 3) were screened for antimalarial activity against *Plasmodium falciparum* [83,84]. The trinuclear complexes **R** and **T** were more active than the dinuclear derivatives **P** and **S** and showed superior activity compared to the metal precursors, suggesting a cooperative effect between the ligand and the metal. Compounds **P**-**T** were screened against the NF54 CQ (chloroquine) sensitive strain of *P. falciparum* and the most active complex **T** was also evaluated against the Dd2 CQ resistant strain [83]. Trinuclear ruthenium complex **T** exhibited potent activity in both the NF54 and Dd2 strains.

Even though no general trend can be identified, several comparative studies of di- and trinuclear vs. mononuclear organometallic complexes have shown a noteworthy increase of cytotoxicity/efficacy with the number of metal units [88,95,96,97]. In this study, we aim to extend the concept of polynuclearity, and describe the synthesis and characterization of conjugates containing two or three covalently inter-connected trithiolato dinuclear ruthenium moieties (see Appendix A for a schematic representation of the compounds). Since complexes with one trithiolato-bridged ruthenium(II)-arene unit were highly cytotoxic against cancer cells [58], and also showed promise as antiparasitic drug candidates [60,61,62], the bioactivity potential of the new conjugates containing two or three di-ruthenium units was also explored and a comparative evaluation of the efficacies is presented.

## 2. Results and discussion

### 2.1. Chemistry

The compounds used in this study not only have a higher charge, but also an important augmentation in size/molecular weight compared to each individual component. Various structural modifications were investigated. In the case of the conjugates with two units, the type of the bonding (ester vs. amide) as well as the nature and length of the linker were systematically varied. In the case of the conjugates containing three units, two topologies were considered: a tripodal star-shape and a linear ‘beads-on-a-string’ arrangement. The nature of the central moiety was also customized (see Appendix A for a schematic representation of the compounds).

Trithiolato-bridged dinuclear ruthenium(II)-arene compounds are stable, which allows the synthesis of more sophisticated molecules by applying ‘chemistry on the complex’. First investigations in this direction were reported in studies on conjugates with peptides [101], the anticancer drug chlorambucil [102] and the fluorophore coumarin [66] designed to improve the water solubility and to promote the anticancer and antiparasitic activities. In general, the synthesis of both the symmetric and mixed trithiolato complexes is straightforward and efficient [57,66].

The mixed intermediate compounds **3**–**5** containing one di-ruthenium unit and one or two hydroxy or amino groups were synthesized following the reactions presented in Scheme 1 by adapting previously described procedures [57,66]. A two-step reaction pathway was used, starting from the [Ru(*η*^6^-*p*-MeC_6_H_4_Pr*^i^*)Cl]_2_Cl_2_ dimer [103]. In the first step, the reaction of two equivalents of (4-(*tert*-butyl)phenyl)methanethiol and, respectively, 4-mercaptophenol with the ruthenium dimer in EtOH (0 °C), under inert atmosphere (N_2_) afforded the symmetric neutral dithiolato intermediates **1** and **2** isolated with 91% yield each. The reactivity of the thiols strongly influences the recovery of the dithioato intermediates, being easily controlled stoichiometrically for **1** when a benzyl thiol was used, but not for **2**, based on a phenyl thiol, for which the respective monothiolato and trithiolato compounds were also formed. The dithiolato intermediates **1** and **2** were further reacted with a third thiol in excess in refluxing EtOH under inert atmosphere (N_2_) to obtain the mixed complexes **3**–**5** isolated in good yields (82%, 84%, and 96%, respectively).

The diester and diamide conjugates containing two di-ruthenium units were obtained by reacting intermediates **3** and **4** with suitably difunctional compounds bearing two acyl chloride or carboxylic acid groups using appropriate reaction conditions (Scheme 2, Scheme 3 and Scheme 4). With few exceptions, the syntheses *per se* did not impose particular issues and the conversions were generally good. Nevertheless, the isolation of the pure compounds proved in some cases less straightforward (yields ranged from 7% to 50%). These purification issues were translated in some cases in poor yields of isolated pure dicationic and tricationic conjugates (see full details in Appendix A). However, the quantities of the isolated products were sufficient for the biological activity screening.

The diester conjugates **6**–**10**, presenting alkyl linkers of different lengths (*n* = 2–6) were obtained with a yield ranging from 7% to 50% by reacting two equivalents of the hydroxy di-ruthenium intermediate **3** with various commercially available bis-acyl chloride compounds in basic conditions (TEA, triethylamine), at low temperature (0 °C) under inert atmosphere (N_2_) (Scheme 3).

The diester conjugates from the reaction of **3** with oxalyl chloride (*n* = 0) and malonyl chloride (*n* = 1) could not be isolated due to the degradation of the acyl chloride reagents in the used reaction conditions. For example, the reaction of **3** with malonyl chloride led to the isolation of the acetyl ester side product.

Similar reaction conditions were used for the synthesis of the alkyl diamide conjugates **11**, **12,** and **14** isolated in 44%, 37%, and 34% yield, respectively. Two equivalents of amino intermediate **4** were reacted with the corresponding bis-acyl chloride compounds (*n* = 0, 4, and 6) in basic conditions (TEA) at low temperature (0 °C) under inert atmosphere. Diamide conjugate **13** (*n* = 5) was obtained with 43% yield using heptanedioic acid, in the presence of HOBt (1-hydroxybenzotriazole) and EDCI (*N*-(3-dimethylaminopropyl)-*N*′-ethylcarbodiimide hydrochloride) as coupling agents and DIPEA (*N*,*N*-diisopropylethylamine) as base (Scheme 3, full experimental details are given in Appendix A).

Diamide conjugates in which the two di-ruthenium units are connected through succinic, malonic and glutaric linkers could not be obtained, neither using the respective bis-acyl chlorides or the corresponding dicarboxylic acids. Degradation of the malonyl chloride reagent was observed in the used reaction conditions. In the cases of the succinyl and glutaryl chlorides, the high reactivity of the di-ruthenium amine **4** favored the formation of the corresponding 5 and 6 atom cyclic amide side products with the 1-pyrrolidine-2,5-dione and 1-piperidine-2,6-dione rings.

The diester **15**–**16** and diamide **17**–**18** conjugates with aromatic linkers were recovered with 32%, 82%, 50%, and 21% yield, respectively using similar conditions, by the reaction of two equivalents of di-ruthenium hydroxy **3** or amino intermediate **4** with commercially available *meta* and *para* substituted bis-acyl chloride compounds in basic conditions (TEA) at low temperature (0 °C) under inert atmosphere (Scheme 4).

For the obtainment of conjugate **20** containing three thiolato-bridged dinuclear ruthenium(II)-arene units in a linear ‘beads-on-a-string’ distribution (Appendix A), two types of ruthenium intermediates were necessary: (i) compound **3** containing one free hydroxy group used for the two capping units, and (ii) the dihydroxy compound **5** (Scheme 1) playing the role of core for connecting two units of **3** via appropriately difunctionalized linkers. Two strategies were considered for the synthesis of this type of conjugates with three di-ruthenium units in a linear arrangement. In one approach, the linkers were anchored on the central dinuclear ruthenium unit in a first step followed by the attachment of the remaining two dinuclear units in a second step. Alternatively, the order of the reactions was reversed, thus first the linkers were reacted with the ‘cap’ ruthenium dinuclear units followed by the connection on the central dinuclear ruthenium moiety. A two-step synthetic pathway was used (Scheme 5 and Scheme 6). First, a glutaric acid linker was attached to the mono-hydroxy di-ruthenium intermediate **3** with the formation of precursor **19** presenting a free carboxylic acid group (Scheme 5). Compound **19** was obtained in 96% yield, by reacting **3** with glutaric anhydride in basic conditions (TEA) at low temperature (0 °C) under inert atmosphere (N_2_).

In the second step, two equivalents of precursor **19** were coupled in the presence of EDCI and DMAP (4-dimethylaminopyridine) with dihydroxy di-ruthenium intermediate **5** to obtain conjugate **20** isolated with 26% yield (Scheme 5).

Two tricationic compounds presenting a star-like architecture were synthesized by attaching three units of the mixed amino di-ruthenium intermediate **4** on two symmetric tripodal core linkers (Appendix A, Scheme 6 and Scheme 7). To minimize steric hindrance for the attachment of the bulky di-ruthenium complexes, central core linkers **21** and **22** with glutaric acid pendant arms were used. These intermediates were obtained with quantitative yield starting from commercially available tris(2-hydroxyethyl)amine, and, respectively, 1,3,5-benzenetrimethanol, following the reactions presented in Scheme 6.

The amide coupling of the core linkers **21** and **22** with mixed amino trithiolato-bridged dinuclear ruthenium(II)*-p*-cymene compound **4** afforded the star-like compounds **23** and **24** (Scheme 7). The reactions were performed in the presence of coupling agents EDCI and HOBt, in basic conditions (DIPEA), under inert atmosphere (N_2_) at room temperature (Scheme 7).

All compounds were characterized by ^1^H and ^13^C NMR spectroscopy, mass spectrometry and elemental analysis (see Materials and Methods—Chemistry section in Appendix A for full details). For diester alkyl linker conjugates **6**–**10**, ester bond formation led to important shifts of the resonances corresponding to the aromatic protons in *α* and *β* position to the former OH group, respectively with Δδ*_H_* ≈ 0.12 ppm highfield, and Δδ*_H_* ≈ 0.41 ppm lowfield. Likewise, the aromatic protons of the *p*-cymene rings were lowfield shifted with about Δδ*_H_* ≈ 0.05–0.26 ppm, and the resonance of the aromatic carbon connected to the ester group (*C*-O-(C=O)) shifted from 160 ppm to 151 ppm.

Similarly, the formation of the amide bond in alkyl connected conjugates **11**–**14** was associated with important shifts of specific resonances in the ^1^H-NMR spectra, which facilitated the monitoring of the reactions’ evolution. The resonance corresponding to the amide hydrogen atom (N*H*-(C=O)) appeared at around 11 ppm. Amide bond formation led to important shifts of the resonances corresponding to the aromatic protons in *α* and *β* position to the former NH_2_ group, respectively with Δδ*_H_* ≈ 1.36 ppm lowfield, and Δδ*_H_* ≈ 0.23 ppm lowfield. Compared to the case of the diester conjugates **6**-**10**, the resonances corresponding to the aromatic protons of the *p*-cymene rings were less affected (lowfield shift Δδ*_H_* = 0.01–0.02 ppm). In conjugates **11**–**14**, the resonance of the aromatic carbon connected to the amide group (*C*-NH-(C=O)) shifts from 149 ppm to 130 ppm.

In the diester (**15**, **16**) and diamide (**17**, **18**) conjugates with aromatic linkers, the resonance of the aromatic protons in *α* and *β* position to the newly formed ester or amide group also shifts compared to corresponding hydroxy and amine di-ruthenium intermediates **3** and **4**. Ester bond formation in **15** and **16** led to shifts of the resonances corresponding to the aromatic protons in *α* and *β* position to the former OH group, with Δδ*_H_* ≈ 0.02 ppm lowfield and, respectively, with Δδ*_H_* ≈ 0.51 ppm lowfield. Amide bond formation in **17** and **18** led to shifts of the resonances corresponding to the aromatic protons in *α* and *β* position to the former NH_2_ group, respectively with Δδ*_H_* ≈ 1.48 ppm lowfield, and Δδ*_H_* ≈ 0.31 ppm lowfield. Nevertheless, the nature of the linker (aliphatic vs. aromatic) had less influence on the NMR resonances compared to the type of bond (ester vs. amide). On the other hand, the resonances of the aromatic carbon connected to the heteroatom of the ester and amide groups (*C*-X-(C=O)) shifted highfield with Δδ*_C_* ≈ 9 ppm for the esters and Δδ*_C_* ≈ 18 ppm for the amides.

For **20**, the tricationic conjugate in which the di-ruthenium units are distributed in a linear arrangement, two sets of resonances could be easily identified corresponding to the two types of thiolato-bridged dinuclear ruthenium(II)-*p*-cymene moieties positioned as core or on the end of the side-arms.

The formation of compound **23** was evidenced by important lowfield shifts of the resonances corresponding to the aromatic protons in *α* and *β* position to the amide (NH-(C=O)) of Δδ*_H_* = 1.31 ppm and Δδ*_H_* = 0.15 ppm, respectively. For the respective carbon atoms in the ^13^C-NMR spectrum of **23** a shift toward low field of Δδ*_C_* = 4.7 ppm and highfield of Δδ*_C_* = 0.8 ppm of the aromatic carbons in *α* and respectively *β* position to the amide (NH-(C=O)) was observed. In addition, an important highfield shift (Δδ*_C_* = 19 ppm) of the aromatic quaternary carbon connected to the nitrogen was also noticed. A similar effect was also observed in the ^1^H-NMR and ^13^C-NMR spectra of triamide **24**.

Electrospray ionization mass spectrometry (ESI-MS) corroborated the spectroscopic data with the intermediates **3**–**5** with one di-ruthenium unit exhibiting molecular ion peaks corresponding to [M-Cl]^+^ ions, conjugates **6**–**18** with two di-ruthenium unit as [M-2Cl]^2+^ ions, and linear and tripodal conjugates **20**, **23,** and **24** with three di-ruthenium units as [M-3Cl]^3+^ ions.

Intermediate complexes **3** and **4** as well as newly obtained polynuclear conjugates **6**–**18** and **20**–**22** were stable in DMSO-d_6_ at 0 °C. (Supplementary Appendix A). Of note, two new very small resonances emerged at ~5.8 and ~8.4 ppm in some spectra recorded after 30 days (for instance for **15**). Yet, although the exact origin of these resonances and the compound(s) they characterize remain unknown, they do not denote a sign of decomposition or formation of a new complex, otherwise new resonances would have appeared in the aromatic region and in the *p*-cymene region around 5.5 ppm and between 0.5 and 3 ppm. Very importantly, recent studies by some of us have shown that dinuclear arene ruthenium complexes are inert to ligand substitutions and remain stable for long period in water solutions or in organic solvents [55,56,57,58].

### 2.2. In Vitro Activity against the Apicomplexan Parasite Toxoplasma gondii

The new conjugates presented in this study were investigated to assess the impact of compound exposure upon *T. gondii* β-gal grown in HFF and non-infected HFF. Cultures exposed to concentrations of 1 and 0.1 µM of each compound of interest (e.g., hydroxy and amino trithiolato-bridged dinuclear ruthenium(II)-arene complexes **3**–**5**, the conjugates with two di-ruthenium units and various types of linkers **6**–**18**, and the conjugates with three di-ruthenium units and different architectures **20**, **23,** and **24**, Appendix A and Figure 4). The results obtained for two other compounds, acetyl ester **25** and amide **26** [104] (see structures in Appendix A) are also presented for comparison. As a measure of parasite proliferation, β-galactosidase activity was determined, while the impact on non-infected HFF was assessed using the AlamarBlue assay.

Compound **5**, possessing two polar hydroxy groups, inhibited *T. gondii* β-gal tachyzoite proliferation more effectively compared to compounds **3** and **4** possessing only one OH or NH_2_ group. Acetyl ester **25** and acetyl amide **26** were also more active against the parasite compared to free OH and NH_2_ analogues **3** and **4**, indicating that the increase in hydrophobicity appears to favor antiparasitic activity. However, **26** is also more cytotoxic against HFF host cells.

In the series of the compounds with two di-ruthenium units and diester alkyl connectors, the derivatives with medium long chain **7** and **8** (*n* = 3 or 4) were active against *T. gondii* β-gal tachyzoites but also exhibited increased toxicity against HFF host cells. From this first screening, the most interesting analogue with an alkyl diester linker was compound **9** (*n* = 5). Diester compounds **6** and **9** were more efficient against *T. gondii* β-gal tachyzoites compared to the hydroxy di-ruthenium compound **3** and exhibited a rather close activity/toxicity profile to acetyl compound **25**.

The diamide derivatives **11**–**14**, with two di-ruthenium units and alkyl linkers, were less cytotoxic in HFF host cells, but displayed also reduced activity with respect to *T. gondii* tachyzoite inhibition compared to the diester compounds **6**–**10**. Derivatives **11**–**14** were also less cytotoxic and less efficacious against *T. gondii* than the acetyl amide **26** or the amino di-ruthenium compound **4**. In this small alkyl diamide set of conjugates, increasing the length of the spacer between the two di-ruthenium units had a favorable effect on the antiparasitic activity.

Interestingly, a similar structural effect (namely amide compounds less active against *T. gondii*/less cytotoxic for HFF) when compared to ester analogues, could be observed in the case of the derivatives **15**–**18** presenting two di-ruthenium units and aryl connectors. Thus, diester compounds **15** and **16** were significantly more active against *T. gondii* tachyzoites compared to their respective diamide analogues **17** and **18**. This might be the consequence of the more important conjugation of the amide bond compared to the ester bond, which can lead to increased rigidity of the molecule, or to the difference of chemical stability of the ester and amide bonds in the cellular environment.

Compound **20** with three di-ruthenium units in a linear distribution displayed an increased activity against *T. gondii* β-gal compared to the two star-shape compounds **23** and **24** but was also more cytotoxic in HFFs. Similar to the di-hydroxy analogue **5** and the acetyl ester **25**, compound **20** showed increased activity against the parasite than the mono-hydroxy di-ruthenium compound **3**. It is important to note that in compound **20** the three di-ruthenium units were connected exclusively through ester bonds, while in compounds **23** and **24** the three units were connected to the central core through amide bonds. The increased molecular volume and rigidity might be responsible for the poor antiparasitic properties of compounds **23** and **24**.

Based on this preliminary screening, twelve compounds exhibiting favorable antiparasitic activity and low or intermediate impairment of HFF viability were selected for the determination of the IC_50_ values in *T. gondii* β-gal and assessment of HFF viability after exposure to 2.5 µM. To determine the IC_50_ value of a compound, two criteria had to be simultaneously satisfied: (i) *T. gondii* β-gal growth was inhibited by 90% or more compared to an untreated control when the compound was applied at 1 µM, and (ii) HFF host cell viability was not impaired by more than 50% for a compound applied at 1 µM. Pyrimethamine, currently used for the treatment of toxoplasmosis, and which inhibited the proliferation of *T. gondii* β-gal tachyzoites with an IC_50_ value of 0.326 µM and did not affect HFF viability at 2.5 µM (Table 1), was used as reference compound. The results are summarized in Table 1, and dose response curves are shown in Appendix A. The selection included the three intermediate compounds **3**, **4**, and **5** containing one di-ruthenium unit and free OH or NH_2_ groups, along with three alkyl diester compounds **6**, **8,** and **9**, the two aryl diester compounds **15** and **16**, as well as compound **20** presenting three di-ruthenium units in a linear arrangement.

Interestingly, the more rigid and sterically hindered aryl diester analogues **15** and **16** were less active in inhibiting *T. gondii* tachyzoite proliferation compared to the alkyl diester compounds **6**, **9** and **10**, and they were also less cytotoxic for HFF at 2.5 µM. Compound **24** with three di-ruthenium units exhibited the highest antiparasitic activity, but was also highly cytotoxic in HFF. Mono- and di-hydroxy compounds **3** and **5** had slightly lower IC_50_ values compared to the amino compound **4**.

The flexibility of the compounds and the molecular volume seemed to have an important impact on the measured IC_50_ values against the apicomplexan parasite *T. gondii*, while the global charge appeared to influence the biological activity to a lesser extent.

### 2.3. In Vitro Anticancer Activity

The cytotoxicity of the di-ruthenium hydroxy and amino intermediates **3**–**5**, of the conjugates with two trithiolato-bridged dinuclear ruthenium(II)-*p*-cymene complexes linked with alkyl diester (**6**, **7,** and **9**), alkyl diamide (**11**–**13**), or aryl linkers (**15**, **17**, **18**), and of the conjugates with three di-ruthenium units **20** and **24** was assessed in the human embryonic kidney cell line HEK293 used as a model for non-cancer cells, in the human ovarian carcinoma cell lines A2780 and A2780cisR, with the latter exhibiting acquired resistance to cisplatin, and in two cell lines A24 and (D-)A24cisPt8.0 that originate from human lung adenocarcinoma (Appendix A, Figure 5 and Figure 6). The A24 cells and its subline (D-)A24cisPt8.0 were described previously [105]. The (D-)A24cisPt8.0 subline is approximately 30-fold more resistant to cisplatin (cisPt) compared to the A24 cells [105]. The cytotoxicity values of cisplatin (cisPt), used as control, are also presented.

Apart from compounds **17** and **18** assessed in A2780 human ovarian cells, all ruthenium compounds were more cytotoxic than cisplatin on all cell lines tested in this study (Figure 5 and Appendix A). In terms of selectivity, compounds **3**–**5** and **11** assessed in cancer cells showed only very reduced selectivity to the non-cancer HEK293 cells. Nevertheless, hydroxy and amine compounds with only one di-ruthenium unit (**3**–**5**) and ester bonded conjugates (**6**, **7**, **9**, **15,** and **20**) present similar cytotoxicity against A2780 and A2780cisR human ovarian cells, and A24 and (D-)A24cisPt8.0 human lung adenocarcinoma cells, irrespectively of the cisplatin resistance profile of the cell lines. This is indicative for a possible different mechanism of action compared to cisplatin, and the potential to circumvent cisplatin cross-resistance. Compounds **3** and **4**, containing only one polar OH and NH_2_ group, respectively, were less cytotoxic on all tested cell lines compared to compound **5** presenting two hydroxy groups.

Irrespective of the nature of the connectors (alkyl or aryl) or of the number of di-ruthenium units (two or three), conjugates with ester bonds presented rather similar antiproliferative profiles, namely they were more cytotoxic compared to conjugates in which the di-ruthenium units were anchored via amide bonds (**11**–**13**, **17**, **18**, and **22**).

Diester compounds **6**, **7**, and **9**, containing two di-ruthenium units and alkyl connectors, were among the most cytotoxic derivatives of the series in all tested cell lines. No or very little selectivity against cancer cells compared to HEK293 cells could be observed for **6**, **7**, and **9**. These compounds appeared to be slightly more cytotoxic on the human lung adenocarcinoma A24 and (D-)A24cisPt8.0 cell lines, compared to the human ovarian cancer cells A2780 and A2780cisR. Almost no influence of the linker length was observed for these conjugates with alkyl diester connectors.

Compounds **11**, **12**, and **13** were less active in the cisplatin resistant cell lines A2780cisR and (D-)A24cisPt8.0 compared to A2780 and A24 cell lines. If these compounds with alkyl diamide linkers were more cytotoxic then cisplatin, they presented lower antiproliferative activities compared to conjugates with alkyl diester connectors **6**, **7**, and **9**. The length increase of the diamide connector was associated with a decrease in cytotoxicity against all tested cancer cells, and this effect was stronger in cisplatin resistant cancer cells A2780 and (D-)A24cisPt8.0.

A difference was perceived between the ester **15** and amide compounds **17** and **18** with two di-ruthenium units and aromatic connectors, the first one being significantly more cytotoxic on all tested cell lines. Thus, diester compound **15** remained among the most cytotoxic of the series in all cell lines (IC_50_ values between 0.036 to 0.094 µM). Diamide compounds **17** and **18** were equally cytotoxic in HEK293 and A2780 cells but were less cytotoxic in the cisplatin resistant A2780cisR cells. Nevertheless, the cytotoxicity profile of human lung adenocarcinoma A24 and (D-)A24cisPt8.0 cancer cells was different from human ovarian cancer cells A2780 and A2780cisR. No important difference was observed between conjugates **6**, **7**, and **9** with alkyl and **15** with aryl diester linkers. The variations were more important in the case of alkyl **11**–**13** and aryl **17**, **18** diamide connected conjugates, especially regarding the antiproliferative activity in the human ovarian cancer cells A2780 and A2780cisR.

Interestingly, noticeable differences in activity could be observed between compounds containing three di-ruthenium units. While compound **20** (with a linear distribution of the three complexes) remains among the most cytotoxic of the series (IC_50_ values ranging between 0.033 and 0.074 µM), the tripodal compound **24** was less cytotoxic (IC_50_ values of 0.167 to 0.277 µM). Whereas in conjugate **20** the three di-ruthenium units are connected via ester bonds, they are anchored to the central core through amide bonds in compound **24**. No important differences in cytotoxicity levels were found in all cancer cells and in HEK293 cells.

Cytotoxicity is a multifactorial process, and resistance to platinum compounds in the A2870cisR cell line is attributed amongst others to decreased uptake, increased glutathione and glutathione S-transferase levels, and increased DNA repair (adduct removal). Uptake may be correlated with Ctr1 transporter expression, which is expressed at lower levels in cisplatin resistant A2870cisR cells compared to sensitive cell lines [106,107,108].

## 3. Materials and Methods

### 3.1. Chemistry

The materials and methods corresponding to the chemistry part are presented with full details in the ‘Materials and methods—Chemistry’ section in Appendix A.

### 3.2. In Vitro Activity Assessment against T. gondii Tachyzoites and HFF

If not otherwise stated, all tissue culture media were purchased from Gibco-BRL, and biochemical agents purchased from Sigma-Aldrich. HFF (SCRC-1041.1) were purchased from ATCC, and were maintained in DMEM medium containing 10% fetal calf serum (FCS, Gibco-BRL, Waltham, MA, USA), and antibiotics as described [109]. Transgenic *T. gondii* β-gal (expressing the β-galactosidase gene from *Escherichia coli*) [110] were kindly provided by Prof. David Sibley (Washington University, St. Louis, MO, USA) and were maintained in HFF as host cells, and were isolated as described before [109].

Compounds were prepared as 1 mM stock solutions in DMSO (dimethyl sulfoxide, Sigma, St. Louis, MO, USA). For activity assays, HFF monolayers were cultured in 96 well plates by seeding 5 × 10^3^ HFF per well and allowing them to grow to confluence in phenol-red free culture medium at 37 °C/5% CO_2_. For infection, *T. gondii* β-gal tachyzoites were separated from their host cells as described [109], parasites were isolated and HFF monolayers were infected with freshly isolated tachyzoites (1 × 10^3^ per well), with compounds solutions added concomitantly during infection.

For the primary screening, *T. gondii* β-gal infected HFF were exposed to 0.1 and 1µM of each compound for a period of 72 h, or the corresponding concentration of DMSO (0.01% and 0.1%, respectively) as control. For the determination of IC_50_ values against *T. gondii* β-gal, compounds were added at 8 concentrations: 0.007, 0.01, 0.03, 0.06, 0.12, 0.25, 0.5, and 1 µM. After 72 h of culture at 37 °C/5% CO_2_, the medium was aspirated, and cells were permeabilized by adding 90 µL PBS (phosphate buffered saline) containing 0.05% Triton X-100. After addition of 10 µL 5 mM chlorophenol red-β-D-galactopyranoside (CPRG; Roche Diagnostics, Rotkreuz, Switzerland) dissolved in PBS, the absorption shift was measured at 570 nm wavelength at various time points using an EnSpire^®^ multimode plate reader (PerkinElmer, Inc, Waltham, MA, USA). For the primary screening at 0.1 and 1 μM, the activity, measured as the release of chlorophenol red over time, was calculated as percentage from the respective DMSO control, which represented 100% of *T. gondii* β-gal growth. For the IC_50_ assays, the activity measured as the release of chlorophenol red over time was proportional to the number of live parasites down to 50 per well as determined in pilot assays. IC_50_ values were calculated after the logit-log-transformation of relative growth and subsequent regression analysis. All calculations were performed using the corresponding software tool contained in the Excel software package (Microsoft, Redmond, WA, USA).

Cytotoxicity assays using uninfected confluent HFF host cells were performed as previously described [111] by AlamarBlue assay. In brief, confluent HFF monolayers in 96 well-plates were exposed to 0.1, 1 and 2.5 µM of each compound. Non-treated HFF as well as DMSO controls (0.01%, 0.1% and 0.25%) were included. After 72 h of incubation at 37 °C/5% CO_2_, the medium was removed, and plates were washed once with PBS. Resazurin stock solution was diluted 1:200 in PBS and 200 µL were added to each well. Plates were read at excitation wavelength 530 nm and emission wavelength 590 nM at the EnSpire^®^ multimode plate reader (PerkinElmer, Inc). Fluorescence was measured at different timepoints. Relative fluorescence units were calculated from timepoints with linear increase.

### 3.3. In Vitro Anticancer Activity

Human ovarian carcinoma cells A2780 and the cisPt resistant variant A2780cisR were purchased from the European Center of Cell Cultures (ECACC, Salisbury, UK). The HEK-293 cell line was kindly provided by Prof. Mühlemann, University of Bern. The A24 cells and its subline (D-)A24cisPt8.0 were described previously [105]. The cancer cells A24 and (D-)A24cisPt8.0 [105] (human lung adenocarcinoma cells wild type A24 and cisPt resistant (D-)A24cisPt8.0 subline), A2780 and A2780cisR (human ovarian cisplatin sensitive and resistant cancer cells), together with HEK293 (human embryonic kidney) cells as model for normal cells, were included in the study for the evaluation of the ruthenium compounds’ selectivity and cross-resistance with cisplatin. The A24 sublines, A2780 and A2780cisR were cultured in RPMI 1640 medium without riboflavin, phenol red and antibiotics, buffered with 4.5 mM HEPES (BioConcept, Allschwil, Switzerland), supplemented with 10% (*v/v*) fetal bovine serum (FBS) (Thermo Scientific, Weltham, MA, USA) as the only source of flavins, and with 13.5 mM NaHCO_3_. In order to retain cisplatin resistance in the A2780cisR cell line, 1 µM cisplatin was added to the medium every 2–3 passages. HEK293 cells were cultured in DMEM/F12 medium, supplemented with 10% (*v*/*v*) FBS. At subcultivation, cell monolayers were rinsed with PBS and exposed for 3 min to StemPro Accutase Cell Dissociation Reagent (Thermo Scientific, Weltham, MA, USA) at 36.5 °C. Detached cells were resuspended in culture medium. Cell densities were determined using Moxi flow cytometer according to manufacturer instructions (Orflo Technologies, Ketchum, ID, USA). Experiments and subcultivations were performed using light with wavelengths above 520 nm to prevent photochemical artifacts. Cytostatic drug response was quantified as previously reported [105]. Briefly, HEK293, A24, A2780, A2780cisR, and (D-)A24cisPt8.0 single cell suspensions were freshly prepared from confluent cell layers using Accutase and diluted in culture medium to a cell density of 2 × 10^4^ or 8 × 10^4^ cells/mL (D-)A24cisPt8.0. 150 µl of this suspension was combined with 150 µl of twice the cisPt or Ru-compound concentration in a 96-well microtiter plate (TPP, Trasadingen, Switzerland). Test plates were incubated under standard conditions for 72 h. Cell densities were determined in a Casy I cell analyzer using Casystat software (OLS OMNI Life Science, Bremen, Germany). Drug concentrations leading to 50% growth inhibition compared to controls were calculated using GraphPad Software (La Jolla, CA, USA). Each experiment was done in triplicates.

## 4. Conclusions

The polynuclear organometallic approach was pioneered by the inorganic trinuclear platinum compound **BBR 3464**. The aim of this study was to extend and apply this concept to new conjugates containing two or three trithiolato-bridged ruthenium(II)-arene units and to assess the biological activity of the new compounds. We synthesized a library of 16 new compounds containing two and three di-ruthenium moieties as well as the corresponding 3 trithiolato di-ruthenium(II)-*p*-cymene intermediates and evaluated the activities of these compounds against T. gondii tachyzoites and cancer cells in vitro.

Remarkably, the in vitro activity assessment against *T. gondii* tachyzoites grown in HFF led to seven compounds (three with one, three with two, and one with three di-ruthenium units), which displayed lower IC_50_ values than the standard drug pyrimethamine. However, the IC_50_ values show that the compounds with two or three trithiolato di-ruthenium(II)arene units do not appear more promising than the intermediates **3**–**5** with only one di-nuclear unit. Of note, the more rigid and sterically hindered aryl diester analogues **15** and **16** with two trithiolato di-ruthenium units are less active against *T. gondii* β-gal and less toxic for HFF host cells compared to the alkyl diester derivatives **6** and **8**–**9**. This insight could be used for further refinement of the structures to improve the activity against *T. gondii*. From these results we conclude that the molecular flexibility and volume of the compounds have an important impact on the measured IC_50_ values against *T. gondii*, while the global positive charge of the compounds influences the biological activity to a lesser extent.

We also determined the cytotoxicity of 14 selected compounds against the cell lines HEK293, A2780, A24, A2780cisR, and (D-)A24cisPt8.0, which resulted in IC_50_ values ranging from 23 to 650 nM. Remarkably, all tested compounds were considerably more cytotoxic than cisplatin in A2780cisR, A24 and de-induced (D-)A24cisPt8.0 subline cancer cells. Compounds **3**–**9**, **11**, **15**, and **20** presented IC_50_ values more than two orders of magnitude lower than those measured for cisplatin against (D-)A24cisPt8.0 cells. Interestingly, irrespective of the nature of the connectors (alkyl or aryl) or of the number or di-ruthenium units (two or three), the conjugates with ester bonds exhibited similar antiproliferative profile, and were more cytotoxic than the conjugates in which the di-ruthenium units were linked via amide bonds. Compounds **3**–**5** with one and **11** with two di-ruthenium units showed a moderate selectivity for the tested cancer cells, and they could be exploited to design new derivatives for future studies. The compounds with two and three di-ruthenium units do not seem more advantageous compared to the intermediate complexes with only one di-nuclear unit **3**–**5**. Nevertheless, the results suggest the potential to circumvent cisplatin cross-resistance.

Overall, this study shows that the concept of extended multinuclearity applied to di-nuclear trithiolato-bridged ruthenium (II)-*p*-cymene compounds is worth additional assessment.

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
