# Peer review of "Conjugates Containing Two and Three Trithiolato-Bridged Dinuclear Ruthenium(II)-Arene Units as In Vitro Antiparasitic and Anticancer Agents"

_pharmaceuticals, 2020, doi:10.3390/ph13120471_

Round 1

Reviewer 1 Report

This manuscript by Studer and colleagues is an interesting work focused on the design, synthesis, characterization and biological evaluation of novel drug conjugates containing two or three trithiolato-bridged dinuclear Ruthenium(II)-arene motifs.

In particular, the authors prepared 24 different Ru(II) complexes and then tested them for their anticancer and antiparasitic activity in vitro, using well-established cell models.

Most of the designed compounds proved to be active showing IC50 values in the low micromolar range, thus stimulating further and deeper research studies.

The target of this manuscript is certainly of great significance and is fully inserted into the promising research field of Ru-based drugs. Indeed, both Ru(II) and Ru(III)-based complexes are attracting increasing attention as second-generation metal-based anticancer agents emerging as valuable alternatives to platinum derivatives in different applications.

Thus, the topic is very attractive and of interest for the typical readership of Pharmaceuticals, giving also an original contribution to this research field.

The experimental part is scientifically accurate and the presented results are sound, also corroborated by a large number of experiments. The work is well organized and comprehensively described.

The paper is logically written and is very easy to read (although there are small typos and grammatical mistakes that need to be fixed before the publication).

In my opinion, some paragraphs are excessively long and some relevant references to previous and pioneering works are missing.

However, I would recommend its publication in Pharmaceuticals provided that a few minor revisions are carried out.

Major points:

  1. A recent review article (Ghosh S. Cisplatin: The first metal based anticancer drug. Bioorg Chem. 2019 Jul;88:102925. doi: 10.1016/j.bioorg.2019.102925.) should be added in the introduction.
  2. Suitable references have to be inserted for the sentence: “Important research efforts are directed towards the development of other platinum(II) or (IV)-based drugs that overcome these drawbacks.” Cite at least some recent papers or in alternative focused review articles.
  3. References relative to Ru(II) and Ru(III) complexes (i.e. 17,28-32) should be extended to other important and pioneering works such as:
  • Zheng, K.;Wu, Q.;Wang, C.; Tan,W.; Mei,W. Ruthenium (II) complexes as potential apoptosis inducers in chemotherapy. Anticancer Agents Med. Chem. 2017, 17, 29–39.
  • Riccardi, C.; Musumeci, D.; Trifuoggi, M.; Irace, C.; Paduano, L.; Montesarchio, D. Anticancer ruthenium (III) complexes and Ru(III) containing nanoformulations: An update on the mechanism of action and biological activity. Pharmaceuticals 2019, 12, 146.
  1. In my opinion, the introduction is very long and should be more concise and better focused.

For example, lines 115-126 can be reduced.

  1. In the Conclusion section, the sentence “Nevertheless, the results suggest different mechanisms of action compared to cisplatin, and the potential to circumvent cisplatin cross-resistance” is not based on the presented results.

A list of minor points to be fixed is here enclosed:

  1. Page 1, line 27: “IC50s

The use of IC50 values in place of IC50s is preferable. Check through the manuscript since it is repeated several times.

  1. Page 1, line 27: “”

Please, indicate in the abstract the use of pyrimethamine as suitable control or reference compound.

  1. Page 1, lines 27-28: “Overall, compound flexibility and volume impacted on the anti-Toxoplasma activity.”

This sentence is not clear. Please, explain better or remove from the abstract.

  1. Page 1, line 28: “The anticancer activity of 14 compounds were…

The anticancer activity…was! The activity is a singular noun.

  1. Page 1, line 31: “In A2780cisR, A24 and (D-)A24cisPt8.0 cells...

Add a comma after cells.

  1. Page 3, line 77: “This opened the door for the development of a myriad of series of organometallic…

of the series…sound better!

  1. Page 4, lines 107-108: “with up to 97% of the Ru content being found in the mitochondrial fraction…

Remove the verb being.

  1. Page 4, line 145: “However, no general trend could be observed…

It is better: no general trend can be determined.

  1. Page 5, lines 157-158: “as well as 1,2-diphenylethylenediamines with controlled configuration

This is an interrupter sentence and should be positioned between commas.

  1. Page 6, line 181: “was shown to dependent on…

Was shown to be dependent is correct.

  1. Page 7, line 225: “The compounds used in this study not only have a higher charge…

Higher compared to??? Comparative adjectives are used to express a comparison between the two terms they refer to.

  1. Page 9, line 277: “…in basic conditions (TEA)…

Please define the acronym TEA for less expert readers.

  1. Page 10, line 316: “…in in basic…

There is a typo error. Remove one “in”.

  1. Page 14, line 446: “In order for a compound to be selected for IC50 determination…

The use of in order for may be wordy. Consider changing the wording or reformatting the sentence.

  1. Table 2, SI: check “Compnds.”.

Author Response

Reviewer 1

This manuscript by Studer and colleagues is an interesting work focused on the design, synthesis, characterization and biological evaluation of novel drug conjugates containing two or three trithiolato-bridged dinuclear Ruthenium(II)-arene motifs.

In particular, the authors prepared 24 different Ru(II) complexes and then tested them for their anticancer and antiparasitic activity in vitro, using well-established cell models.

Most of the designed compounds proved to be active showing IC50 values in the low micromolar range, thus stimulating further and deeper research studies.

The target of this manuscript is certainly of great significance and is fully inserted into the promising research field of Ru-based drugs. Indeed, both Ru(II) and Ru(III)-based complexes are attracting increasing attention as second-generation metal-based anticancer agents emerging as valuable alternatives to platinum derivatives in different applications.

Thus, the topic is very attractive and of interest for the typical readership of Pharmaceuticals, giving also an original contribution to this research field.

The experimental part is scientifically accurate and the presented results are sound, also corroborated by a large number of experiments. The work is well organized and comprehensively described.

The paper is logically written and is very easy to read (although there are small typos and grammatical mistakes that need to be fixed before the publication).

In my opinion, some paragraphs are excessively long and some relevant references to previous and pioneering works are missing.

However, I would recommend its publication in Pharmaceuticals provided that a few minor revisions are carried out.

  • We thank this referee for his positive comments, the careful readings and the numerous suggestions that help us for improving our manuscript.

Major points:

  1. A recent review article (Ghosh S. Cisplatin: The first metal based anticancer drug. Bioorg Chem. 2019 Jul;88:102925. doi: 10.1016/j.bioorg.2019.102925.) should be added in the introduction.

  • This reference has been added

  1. Suitable references have to be inserted for the sentence: “Important research efforts are directed towards the development of other platinum(II) or (IV)-based drugs that overcome these drawbacks.” Cite at least some recent papers or in alternative focused review articles.

  • We have added additional references

  1. References relative to Ru(II) and Ru(III) complexes (i.e. 17,28-32) should be extended to other important and pioneering works such as:
  • Zheng, K.;Wu, Q.;Wang, C.; Tan,W.; Mei,W. Ruthenium (II) complexes as potential apoptosis inducers in chemotherapy. Anticancer Agents Med. Chem. 2017, 17, 29–39.
  • Riccardi, C.; Musumeci, D.; Trifuoggi, M.; Irace, C.; Paduano, L.; Montesarchio, D. Anticancer ruthenium (III) complexes and Ru(III) containing nanoformulations: An update on the mechanism of action and biological activity. Pharmaceuticals 2019, 12, 146.

  • These references have been added

  1. In my opinion, the introduction is very long and should be more concise and better focused. For example, lines 115-126 can be reduced.

  • We agree that our introduction is rather long, although from about 4 pages, 1 ½ page is devoted to figures. The text of the introduction thus represents about 2 ½ pages for a total manuscript length of 27 pages, which is in our opinion still acceptable. We have to mention that in this work, we report the synthesis of new ruthenium conjugates made of one, two or three arene ruthenium subunits, and their in vitro properties against cancer cells and parasites. All together, these aspects require to mention and describe a lot of previous work, from various fields like Ruthenium compounds as anticancer agents, as antiparasitic compounds, the concept of multinuclearity applied to organometallic drugs.
  • We have tried to be more focused and have slightly shorten the text wherever it was in our opinion possible.

  1. In the Conclusion section, the sentence “Nevertheless, the results suggest different mechanisms of action compared to cisplatin, and the potential to circumvent cisplatin cross-resistance” is not based on the presented results.

  • We agree with the referee and we have removed this part of the sentence in the revised manuscript.

A list of minor points to be fixed is here enclosed:

  1. Page 1, line 27: “IC50s”

The use of IC50 values in place of IC50s is preferable. Check through the manuscript since it is repeated several times.

  • This was done

  1. Page 1, line 27: “”

Please, indicate in the abstract the use of pyrimethamine as suitable control or reference compound.

  • We have modified this sentence accordingly

  1. Page 1, lines 27-28: “Overall, compound flexibility and volume impacted on the anti-Toxoplasma activity.”

This sentence is not clear. Please, explain better or remove from the abstract.

  • Volume was replaced by size. We hope the sentence is better understandable now.

  1. Page 1, line 28: “The anticancer activity of 14 compounds were…”

The anticancer activity…was! The activity is a singular noun.

  • Corrected

  1. Page 1, line 31: “In A2780cisR, A24 and (D-)A24cisPt8.0 cells...”

Add a comma after cells.

  • Done

  1. Page 3, line 77: “This opened the door for the development of a myriad of series of organometallic…”

of the series…sound better!

  • We actually prefer to let myriad, to highlight that as really a lot of compounds have been developed!

  1. Page 4, lines 107-108: “with up to 97% of the Ru content being found in the mitochondrial fraction…”

Remove the verb being.

  • Done

  1. Page 4, line 145: “However, no general trend could be observed…”

It is better: no general trend can be determined.

  • Changed

  1. Page 5, lines 157-158: “as well as 1,2-diphenylethylenediamines with controlled configuration”

This is an interrupter sentence and should be positioned between commas.

  • Done

  1. Page 6, line 181: “was shown to dependent on…”

Was shown to be dependent is correct.

  •  

  1. Page 7, line 225: “The compounds used in this study not only have a higher charge…”

Higher compared to??? Comparative adjectives are used to express a comparison between the two terms they refer to.

  • This has been corrected. …weight compared to singular unit complexes.

  1. Page 9, line 277: “…in basic conditions (TEA)…”

Please define the acronym TEA for less expert readers.

  • The acronym TEA is already described a couple of lines above, p9, line 267

  1. Page 10, line 316: “…in in basic…”

There is a typo error. Remove one “in”.

  • Done

  1. Page 14, line 446: “In order for a compound to be selected for IC50 determination…”

The use of in order for may be wordy. Consider changing the wording or reformatting the sentence.

  • Done: “For a compound to be selected for IC50 determination, two criteria had to be simultaneously satisfied”

  1. Table 2, SI: check “Compnds.”.

  • Replaced by compounds

Reviewer 2 Report

This manuscript deals with an interesting design of ruthenium complexes based on the polynuclearity other successful platinum complexes as anticancer drug.

The chemistry seems to have been performed properly though there is a lack of analysis data in some of the compounds with only mass spectra, this last not being an absolute analytical tool. Analysis data should be added.

Regarding the study of the complexes, those are extremely active in cancer cells but also in the non-cancer model: embryonic kidney 293 cells. So I don’t see those are good models, might be embryonic cells are not the most appopiated choice. There are some cases where the ic50 is hgh but it is also high in cancer cells, so they might be a better option as antiparasitic compoudns.

In general both options seems to be very different and the introduction of the design and antiparasitic mechanism is very poor, or absent.

Said this, there is a very important part of the study missing in this manuscript: the analysis of the compound in solution. The spectra and studies of the compound in solutions should be monitored over the time to prove stability (UV in tris buffer, Hplc, or NMR…). The integrity of the compound versus solvents (dmso.. so on) and buffer ions (water, tris..) is a mandatory study with metallodrugs.

Author Response

"This manuscript deals with an interesting design of ruthenium complexes based on the polynuclearity other successful platinum complexes as anticancer drug.

  • We thank this referee for this nice comment.

The chemistry seems to have been performed properly though there is a lack of analysis data in some of the compounds with only mass spectra, this last not being an absolute analytical tool. Analysis data should be added.

  • We do not understand the criticism of this referee on this point: Indeed, all compounds have been analyzed using NMR and most of them also using EA in addition to MS data (all data are provided in the SI).

Regarding the study of the complexes, those are extremely active in cancer cells but also in the non-cancer model: embryonic kidney 293 cells. So I don’t see those are good models, might be embryonic cells are not the most appopiated choice. There are some cases where the ic50 is hgh but it is also high in cancer cells, so they might be a better option as antiparasitic compoudns.

  • In the light of the obtained results, Probably. Although cisplatin does not really perform better (in terms of selectivity towards HEK cells) than some of the compounds presented in this work

In general both options seems to be very different and the introduction of the design and antiparasitic mechanism is very poor, or absent.

  • We feel on the contrary that the choice and the design of our compounds are well described and justified in the introduction (which was by the way judged as a bit too long regarding this aspect by referees 1&3).
  • The understanding of the antiparasitic mechanism of these compounds does not represent the topic of this manuscript.

Said this, there is a very important part of the study missing in this manuscript: the analysis of the compound in solution. The spectra and studies of the compound in solutions should be monitored over the time to prove stability (UV in tris buffer, Hplc, or NMR…). The integrity of the compound versus solvents (dmso.. so on) and buffer ions (water, tris..) is a mandatory study with metallodrugs."

  • Such stability tests have been performed and are presented in the SI (Figures S4 to S8). The stability of thiolato-bridged Ru compounds in water and buffered solutions has been long established (see Refs 55 to 58). The reason is that, unlike many other arene ruthenium compounds, these complexes do not have a labile, hydrolysable Ru-Cl bond. We have added this sentence in the text: “Very importantly, recent studies by some of us have shown that dinuclear arene ruthenium complexes are inert to ligand substitutions and remain stable for long period in water solutions or in organic solvents.” Lines 394-396.

Reviewer 3 Report

The paper submitted for publication in “Pharmaceuticals” by V Studer et al. is related to the synthesis, characterization and antiparasitic/antiproliferative effects of some ruthenium(II)-arene complexes. The authors use one of the most studied classes of Ru (II) organometallic compounds, as "construction unit" in the synthesis of a class of compounds that is still relatively little studied. In addition, an impressive number of new structurally related compounds has been developed and studied in this work. It is the reviewer’s opinion that several issues have to be addressed to improve the manuscript’s quality and its suitability for publication in high impact journal such as Pharmaceuticals.
The Introduction needs to be rewritten, considering the following points:
  1. line 48, the sentence “Currently, there are six platinum drugs with marketing approval…” does not make sense here, since it is only at the end of this paragraph that the authors talk about platinum drugs
  2. line 65, the authors state that “Among the numerous organometallics presenting anticancer properties, ruthenium(II) and (III) complexes were identified as…”, what kind of compounds are the authors referring to? Because then the authors present 3 Ru (III) complexes (NAMI-A, KP1019 and TLD1433) that are inorganic. Moreover, throughout the manuscript many inorganic compounds are wrongly classified as organometallic
  3. line 67, the sentence “Ruthenium oxidation state, as well as compound solubility…” must be clarified
  4. line 85, in the same way that the authors justify the choice of this type of compounds for cancer would be interesting to support the choice as antiparasitic
  5. line 137, the platinum compound BBR 3464 it is not organometallic, it is inorganic
  6. line 144, this paragraph should be substantiated with references
  7. The introduction referring to Ru-arene compounds it is excessively descriptive. I understand that the authors want to explain the effects observed for each substituent (and this is important), but both schemes and text can be more synthetic.
  8. The numbering of the figures and tables in the Supporting Information (SI) is wrong.
  9. Figure S1 with the general structure of the conjugates to be synthesized should be in the main text and not in the SI. If the authors consider that they have too many figures, they can place in SI the schemes of synthesis of the compounds that have already been published.
  10. Schemes 5 and 6, since they correspond to the same synthesis should be merged into one.
  11. In scheme 8 it would be interesting to have the complete structure of complexes 23 and 24.
  12. Figures S4 to S8 are not practical to read. Since what is important to observe in these figures is the existence (or not) of differences in the complexes right after the sample preparation and after >30 days storage at 0-5°C in the dark, it would be more visible to the reader if the two spectra of the same compound were together.
The authors state that the compounds are all stable at the end of this time, however for some compounds the formation of a new compound begins to be observed (example compound 15).
  1. The previous studies only give an indication if the compound structure changes frozen ate 05ºC, however for a biological application it is very important to understand what happens to the compound in an aqueous or cellular medium.
Is this compound water-soluble or does it need a co-solvent (probably DMSO)?
In the experimental part of the IC50 determination it is not clear what the procedure is like, what the incubations are like, etc.
Authors must determine stability over time until midnight (which they use for cell studies) in the co-solvent they use and, in the co-solvent/cellular medium. This point is crucial before any biological investigation.
  1. the paragraph that starts at line 545 does not make sense, since they are unpublished data and do not bring anything new to this manuscript.
  2. The experimental part should come in the manuscript and not in the SI.
  3. Experimental section: the NMR assignment is extremely confusing, the authors must number each atom in the complex schemes (1, 2, 3 ...) and use these numbers in the atribution of each H and C in the experimental part.
  4. Experimental section: Compounds 2 and 5 are not new and the reference of the procedure used must be clearly indicated in the experimental part
  5. Experimental section: for some compounds the authors do not show results of elementary analyses. Is there any explanation for this? These data must be provided.
For other compounds, such as 9, 12 and14, are elementary analyses correct with a very large amount of solvent? Which is strange, since these solvents leave with drying prior to the study. Authors should clarify this point by repeating the elementary analyses.
  1. Conclusions: BBR3464 is inorganic!

Author Response

Reviewer 3

The paper submitted for publication in “Pharmaceuticals” by V Studer et al. is related to the synthesis, characterization and antiparasitic/antiproliferative effects of some ruthenium(II)-arene complexes. The authors use one of the most studied classes of Ru (II) organometallic compounds, as "construction unit" in the synthesis of a class of compounds that is still relatively little studied. In addition, an impressive number of new structurally related compounds has been developed and studied in this work. It is the reviewer’s opinion that several issues have to be addressed to improve the manuscript’s quality and its suitability for publication in high impact journal such as Pharmaceuticals.

  • We thank this referee for his positive comments, the careful readings and the numerous suggestions that help us for improving our manuscript.

The Introduction needs to be rewritten, considering the following points:

  1. line 48, the sentence “Currently, there are six platinum drugs with marketing approval…” does not make sense here, since it is only at the end of this paragraph that the authors talk about platinum drugs.

  • We agree, and have moved this sentence.

  1. line 65, the authors state that “Among the numerous organometallics presenting anticancer properties, ruthenium(II) and (III) complexes were identified as…”, what kind of compounds are the authors referring to? Because then the authors present 3 Ru (III) complexes (NAMI-A, KP1019 and TLD1433) that are inorganic. Moreover, throughout the manuscript many inorganic compounds are wrongly classified as organometallic.

  • We agree and have corrected this wrong classification of compounds throughout the manuscript. Organometallic compounds are presented after this sentence.

  1. line 67, the sentence “Ruthenium oxidation state, as well as compound solubility…” must be clarified

  • We have modified this sentence. The new sentence is: For these ruthenium compounds, the oxidation state of the Ruthenium atom, their solubility and stability in physiological media greatly influence their activity against cancer cells.

  1. line 85, in the same way that the authors justify the choice of this type of compounds for cancer would be interesting to support the choice as antiparasitic

  • This is now provided in lines 111-129. Additionally, we have added this sentence “These encouraging results stimulated us, and we decided to further develop and evaluate new libraries of complexes based on the trithiolato-bridged di-ruthenium scaffold as antiparasitic agents” line 132-134.

  1. line 137, the platinum compound BBR 3464 it is not organometallic, it is inorganic

  • Absolutely! Corrected.

  1. line 144, this paragraph should be substantiated with references

  • We have added references 88, 92, 95, 98.

  1. The introduction referring to Ru-arene compounds it is excessively descriptive. I understand that the authors want to explain the effects observed for each substituent (and this is important), but both schemes and text can be more synthetic.

  • It’s true that this part is descriptive and may be long. However, we think that it is necessary to keep all the details provided, as a lot of previous work, encompassing a lot of research fields like Ruthenium compounds as anticancer agents and as antiparasitic compounds, the concept of multinuclearity applied to organometallic drugs, that must be mentioned and the main findings exposed in the introduction.

  1. The numbering of the figures and tables in the Supporting Information (SI) is wrong.

  • We thank this referee for having noticed this. This was corrected.

  1. Figure S1 with the general structure of the conjugates to be synthesized should be in the main text and not in the SI. If the authors consider that they have too many figures, they can place in SI the schemes of synthesis of the compounds that have already been published.

  • This was our choice, as the chemical structure of all compounds are shown subsequently in the Synthesis section. We’d prefer to let this Figure as Figure S1, but it could be of course moved in the main text if absolutely required.

  1. Schemes 5 and 6, since they correspond to the same synthesis should be merged into one.

  • We agree, this has been done.

  1. In scheme 8 it would be interesting to have the complete structure of complexes 23 and 24.

  • The complete structures of complexes 23&24 are very large, and even as a 2 column figure, the labels and bonds are relatively small. We have therefore chosen to represent them as currently shown in scheme 8, which is in our opinion the best option.

  1. Figures S4 to S8 are not practical to read. Since what is important to observe in these figures is the existence (or not) of differences in the complexes right after the sample preparation and after >30 days storage at 0-5°C in the dark, it would be more visible to the reader if the two spectra of the same compound were together.

  • In principle, yes, but since the spectra of each individual compound are identical after sample preparation and after 30 days, we believe that these figures can remain in this form without detriment for the reading/evaluation. In the manuscript, the figures may also be placed face-to-face, the comparison for 1 complex being direct and easy.

The authors state that the compounds are all stable at the end of this time, however for some compounds the formation of a new compound begins to be observed (example compound 15).

  • For 15, there are indeed 2 new very small resonances that emerge at ~5.8 and ~8.4 ppm in the spectrum recorded after 30 days. Yet, although the exact origin of these resonances and the compound(s) they characterize remain unknown, we believe that they do not denote a sign of decomposition or formation of a new complex. Indeed, decomposition or formation of a new complex are clearly visible in the aromatic region and in the p-cymene region around 5.5 ppm and between 0.5 and 3 ppm. For all complexes, clearly, no new resonances or a drop of the resonance’s intensity can be noticed in these regions.

We also would like to point out that 30 days exceeds by far the stability required for metallodrugs, which is usually 72h.

  1. The previous studies only give an indication if the compound structure changes frozen ate 05ºC, however for a biological application it is very important to understand what happens to the compound in an aqueous or cellular medium.

  • The stability of thiolato-bridged Ru compounds in water and buffered solutions has been long established (see Refs 55 to 58). The reason is that, unlike many other arene ruthenium compounds, these complexes do not possess a labile, hydrolysable Ru-Cl bond.

Is this compound water-soluble or does it need a co-solvent (probably DMSO)?

  • The compounds are only very sparingly soluble in water (less than 0.5 mg / ml)

DMSO was used as co-solvent and this is described in the biological part of the manuscript.

In the experimental part of the IC50 determination it is not clear what the procedure is like, what the incubations are like, etc.

  • Details have been added.

Authors must determine stability over time until midnight (which they use for cell studies) in the co-solvent they use and, in the co-solvent/cellular medium. This point is crucial before any biological investigation.

  • The stability of thiolato-bridged Ru compounds in water and buffered solutions has been long established (see Refs 55 to 58). The reason is that, unlike many other arene ruthenium compounds, these complexes do not possess a labile, hydrolysable Ru-Cl bond. “Very importantly, recent studies by some of us have shown that dinuclear arene ruthenium complexes are inert to ligand substitutions and remain stable for long period in water solutions or in organic solvents.” Lines 394-396.

  1. the paragraph that starts at line 545 does not make sense, since they are unpublished data and do not bring anything new to this manuscript.

  • We agree and have removed this paragraph

  1. The experimental part should come in the manuscript and not in the SI.

  • This was again our choice, especially because this experimental part is excessively long. Inserted in the manuscript, the exp part would lengthen it considerably, the manuscript already being 27 pages!. Therefore, we’d prefer to let the exp part in the SI, but it could be of course moved in the main text if absolutely required.

  1. Experimental section: the NMR assignment is extremely confusing, the authors must number each atom in the complex schemes (1, 2, 3 ...) and use these numbers in the atribution of each H and C in the experimental part.

  • This notation is the one used by us in all previous reports related to these ruthenium compounds, and we do not think it is confusing. Furthermore, this notation/assignment is very often used for describing new compounds in similar manuscripts, also in Pharmaceuticals (for instance in Pharmaceuticals 2020, 13(11), 393, https://doi.org/10.3390/ph13110404 Pharmaceuticals 2020, 13(11), 404, https://doi.org/10.3390/ph13110393 Pharmaceuticals 2020, 13(11), 405, https://doi.org/10.3390/ph13110405, Pharmaceuticals 2020, 13(11), 391; https://doi.org/10.3390/ph13110391

If absolutely necessary, we could of course number each atom in the schemes and use them for attributing each H and C in the NMR spectra, but this is a tedious and very time consuming task that in our opinion does not provide new information.

  1. Experimental section: Compounds 2 and 5 are not new and the reference of the procedure used must be clearly indicated in the experimental part.

  • We agree and added the procedure and relevant references.

  1. Experimental section: for some compounds the authors do not show results of elementary analyses. Is there any explanation for this? These data must be provided.

For other compounds, such as 9, 12 and 14, are elementary analyses correct with a very large amount of solvent? Which is strange, since these solvents leave with drying prior to the study. Authors should clarify this point by repeating the elementary analyses.

  • We have measured or re-measured EA for some of the compounds for which these data were missing or not of high quality (compound 6, 9, 11, 12, 14, 16, 17, 18, 20. For compounds 21 & 22, consistency (viscous oil) prevents EA.

The presence of water can be explained from the storage in the fridge for 24 -48 h before the EA measurement. (hint form our EA team, not only for these compounds).

  1. Conclusions: BBR3464 is inorganic!

  • Absolutely, corrected.

Round 2

Reviewer 3 Report

I think the article is much better and ready to be published, I just suggest that the authors add some notes about the impurities of complex 15, and why they don't have EA for complexes 21 and 22.

Author Response

I think the article is much better and ready to be published, I just suggest that the authors add some notes about the impurities of complex 15, and why they don't have EA for complexes 21 and 22.

We thank this referee for his/her positive evaluation. We have added the suggested notes in our manuscript.